statistical physics/systems theory/artificial intelligence

data science, terrorism, conflict, forecasting models

**Author for correspondence:**
Weisi Guo
e-mail: wguo@turing.ac.uk

# Common statistical patterns in urban terrorism

Weisi Guo[1,2]

[1]The Alan Turing Institute, London, UK
[2]Warwick Institute for Science of Cities, University of Warwick, Coventry, UK

WG, 0000-0003-3524-3953

The underlying reasons behind modern terrorism are seemingly complex and intangible. Despite diverse causal mechanisms, research has shown that there exists general statistical patterns at the global scale that can shed light on human confrontation behaviour. While many policing and counter-terrorism operations are conducted at a city level, there has been a lack of research in building city-level resolution prediction engines based on statistical patterns. For the first time, the paper shows that there exist general commonalities between global cities under frequent terrorist attacks. By examining over 30 000 geo-tagged terrorism acts over 7000 cities worldwide from 2002 to today, the results show the following. All cities experience attacks $A$ that are uncorrelated to the population and separated by a time interval $t$ that is negative exponentially distributed with a death-toll per attack that follows a power-law distribution. The prediction parameters yield a high confidence of explaining up to 87% of the variations in frequency and 89% in the death-toll data. These findings show that the aggregate statistical behaviour of terror attacks are seemingly random and memoryless for all global cities. They enabled the author to develop a data-driven city-specific prediction system, and we quantify its information-theoretic uncertainty and information loss. Further analysis shows that there appears to be an increase in the uncertainty over the predictability of attacks, challenging our ability to develop effective counter-terrorism capabilities.

## 1. Introduction

Understanding complex human interactions is vital for solving some of humanity's most pressing social challenges [1]. One of these challenges is the protracted political violence that plagues many urban regions in the world [2]. While creating data-driven regression models can yield insights into ongoing violence [3,4], statistical patterns can also yield insight into common trends [5,6]. In this paper, we focus our analysis on urban attacks, where the majority of attacks take place (e.g. for casualty and impact maximization [7,8]), but note that rural under-reporting is an open challenge.

Statistical analysis of complex processes, even across diverse genres and mechanisms have value in data-driven prediction. It has been shown that many complex processes with a multitude of different causal factors can exhibit common statistical patterns that aid prediction, e.g. bus arrival time in busy urban areas.

## 1.1. Review of statistical analysis

The science of finding patterns in war stretch back to the 1940s, when Richardson showed that the intensity of major battles in the Victorian era fits power-law distributions [9–11]. This has been reinforced for conflict and terrorism data in the modern era [5,6,12]. Tracking trends in both large-scale wars and regional political violence is important in quantifying the effectiveness of peacekeeping and peace negotiation efforts [1,5,13–15].

In terms of temporal analysis (frequency or time interval between attacks), the frequency of large terrorist attacks has also been shown to obey a power-law distribution [16,17]. Regional diffusion of violence has been modelled by point processes with a stochastic integral kernel [18,19]. Small-scale local events have also been studied, i.e. improvised explosive device (IED) attacks have been shown to exhibit self-excitation behaviour modelled by a Hawkes process [20]. Long-term trend analysis has been conducted recently [21], whereby it is argued that our current period of relative peace from major wars is statistically insignificant.

Detailed causal mechanisms on why certain locations experience more conflict or longer duration conflict have been studied [22]. For example, regions far from government control and rich in natural resources tend to experience protracted conflict [23–25]. However, not all conflict is driven by such mechanisms, e.g. protracted urban warfare in Colombian cities. It has also been shown that it is difficult to separate the different genres of conflict, e.g. civil war and terrorism [26–28] and as such it does make sense to consider them together from a statistical modelling or prediction perspective.

The majority of statistical studies are still focused on one of the following: (i) aggregate scale attacks across either a large region or the whole world [5,6,9,10,12,16,17,21,29,30], (ii) highly scenario/conflict specific violence, often with a specific violence genre (i.e. IED attacks in Northern Ireland [20], Afghan war [18], ethnolinguistic tensions [31–33] and natural resources driven conflict [23]), or (iii) long-term historical trends [11,34] that span several centuries and are associated with other temporal factors (e.g. climate change [35–37]). It remains an open question whether each city experiences *a common human ecological behaviour in the frequency and size of attacks.* If so, prediction engines [1,3] would inform urban policing and counter-terrorism policies and lead to *city-specific prediction engines.*

## 1.2. Contribution

Recent attempts have examined general behaviour at national statistical levels [38] and across different confrontation genres [39]. However, detailed geographical analysis (city scale) across all genres and geographies is lacking. Indeed, city-scale modelling is important as counter-terrorism policies are often adopted at the city scale (i.e. London and New York suffer disproportionately more threats and attacks than other cities) [40]. Understanding a common ecological behaviour at detailed city resolution can help stakeholders to create models and make forecasts.

This paper sets out to do this. In this paper, the author offers insight that inter-relates the intensity, frequency and prediction uncertainty of terrorist attacks in different cities worldwide since 2001. The results across different *urban ecologies* show that the *intensity* (death-toll per attack) data still obey a power law [16,17], the *frequency* (interval between attacks) is exponentially distributed. Here, we show that despite diverse conflict genres and multiple confounding mechanisms in play, all global cities suffer attacks describable by a common statistical pattern. The memoryless nature of this pattern suggests that multiple causal mechanisms are independent to each other and that prediction is not helped by the knowledge of previous attacks.

This enables us to build a simple city-specific predictor based on past attack data, and we show both the information theoretic uncertainty and information loss in attempting to predict the underlying terrorism process. Finally, we use spectrogram analysis to further show that there is a growing uncertainty hidden in the complex process.

## 2. Results

The geo-tagged terrorism and unconventional conflict data from the Global Terrorism Database (GTD) [41] were analysed. All data used are available in the Dryad Digital Repository [42]. Since post-Cold War, violence between terrorism, politics, criminal enterpriser (e.g. narcotics) has become interleaved.

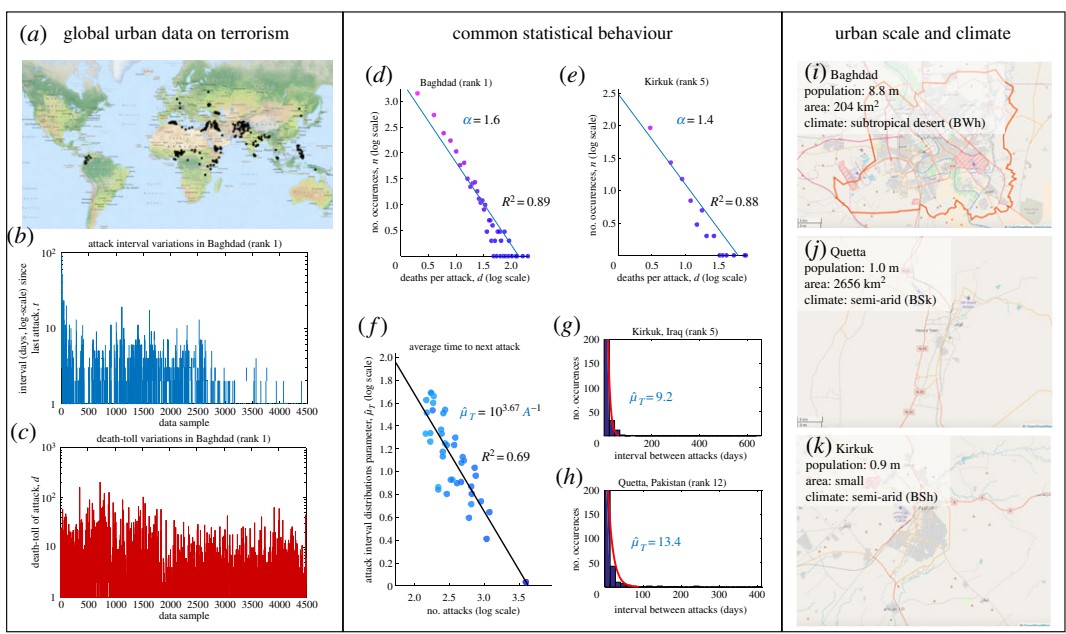

**Figure 1.** Terrorist attack intensity and interval in top 40 conflict cities with diverse urban scales and climates: (a) Map of terrorist incidents (2002–2014)—the black stars indicate the top 150 conflict locations (clustered to nearest city) with highest aggregate death-tolls. (b,c) Example of data in the highest attacked city (Baghdad) showing variations in death-toll and frequency as a function of time (day count). (d,e) The attack intensity follows power-law distribution with exponent parameters $\alpha$ for two example cities. (f) The predicted average attack interval parameter $\hat{\mu}_T$ is linearly correlated with the average number of attacks $\hat{\mu}_T = 10^{3.67}/A$, where $10^{3.67}$ is the number of days in the recent 13 year interval. The actual number of attacks in each city explains for 69% of the variation (adjusted $R^2 = 0.69$) in the prediction parameter for all cities. Panel (g,h) shows two example cities and the negative exponential distribution fit for interval (days) between attacks alongside the estimated parameter $\hat{\mu}_T$. Panel (i–k) shows three example cities and their diverse population size (varies by one order of magnitude), diverse city area (varies by one order of magnitude) and different climates.

Often, trans-national organizations like ISIS participate in all above aspects. As such, studies have shown that it has become difficult to separate the different genres of violence both statistically and contextually [26]. Therefore, it makes sense to consider GTD in its entirety, which is the violence between a non-state actor and other targets (state or non-state).

The results show that the vast majority of conflict incidents occur in close proximity to an urban area with a mean distance of 27 km. This highlights the importance of focusing on city-/town-scale resolution analysis. Figure 1a shows a map of terrorist incidents (2002–2014), where the black stars indicate the top 150 conflict locations (clustered to the nearest city) with highest aggregate death-tolls. Figure 1b,c shows example of data in the highest attacked city (Baghdad) with variations in death-toll and frequency as a function of time (day count). As the data are across the whole world, a significant portion of that data are not 'War or Terror' related (e.g. Colombia, Narco-War Mexico, political violence in India).

## 2.1. Intensity: power-law distribution

The results in figure 1d,e show that the terrorist intensity (death-toll per attack) is distributed in accordance with the established power-law distribution [6,10,12]. The exponent parameters $\alpha$ for two random cities are presented. The interesting observation is that most previous studies have considered low-resolution conflicts (major wars) that span over 100 years or sub-national regional studies [43,44], and it seems that the power-law distribution remains valid even for high-resolution terrorism and non-conventional conflict data in the modern era. What is less understood is how the time interval between attacks is distributed, and this is the focus of the paper.

## 2.2. Interval: negative exponential distribution

The results show that the time interval between sequential attacks $t$ fits the negative exponential distribution. The probability density function (PDF) of a negative exponential distributed variable $t$ with support $[0, +\infty)$ is:

$$f(t; \hat{\mu}_T) = \frac{1}{\hat{\mu}_T} \exp\left(\frac{-t}{\hat{\mu}_T}\right),$$

(2.1)

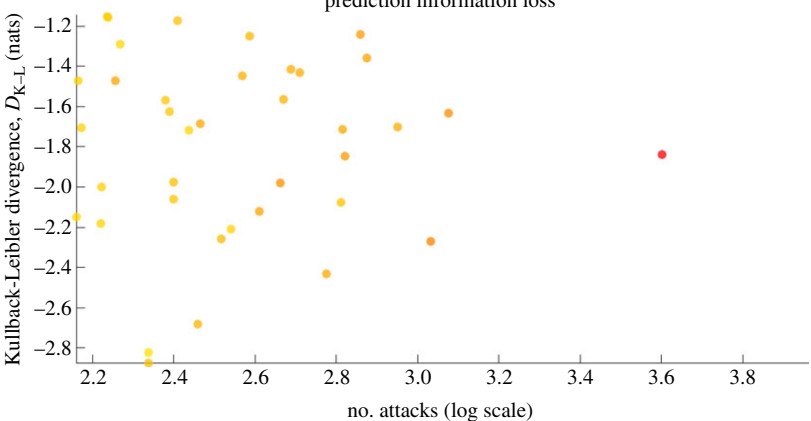

**Figure 2.** Accuracy and information loss in prediction: directed K–L divergence (information loss) for predicting the time interval between attacks. Colour indicates the number of casualties (red is highest).

where the parameter $\hat{\mu}_T$ is the distribution parameter. Note, the exponential decay rate is given by $1/\hat{\mu}_T$ and the variance is given by $\hat{\mu}_T^2$. Indeed, there are prior work to support this for civil war models that can be modelled using zero-inflated count models [45].

Figure 1$f$–$h$ shows the terrorist attack interval in the top 40 conflict cities. In general, all cities examined experience attacks that are separated by a time interval ($t$, days) that is negative exponentially distributed $\sim \exp(-\hat{\mu}_T)$. The results show two example cities and the negative exponential distribution fit for interval between attacks and the deaths per attack, alongside the estimated distribution parameters $\hat{\mu}_T$. Under the maximum-likelihood (ML) estimator, the exponential distribution's parameter $\hat{\mu}_T$ is equal to the mean of the data $\mu$, i.e. $\hat{\mu}_T = \sum_i T_i/A$, where $T_i$ is the actual interval between any two attacks, and $A$ is the total number of attacks in the city over all time ($10^{3.67}$ days in 2002–2014). Figure 1$f$ shows the predicted interval parameter $\hat{\mu}_T$ is linearly correlated with the average number of attacks $\hat{\mu}_T = 10^{3.67}/A$. The actual number of attacks in each city explains for 69% of the variation (adjusted $R^2 = 0.69$) in the distribution parameter $\hat{\mu}_T$.

For a historical data sample of terrorist attacks that is $n$ in size, the lower- and upper-bound of the exponential distribution parameter is given as

$$\hat{\mu}_{T,\text{upper}} = \hat{\mu}_T \left(1 - \frac{1.96}{\sqrt{n}}\right)^{-1} \text{and} \quad \hat{\mu}_{T,\text{lower}} = \hat{\mu}_T \left(1 + \frac{1.96}{\sqrt{n}}\right)^{-1}. \tag{2.2}$$

For the top 40 cities considered in the analysis, the number of attacks in 2002–2014 is between 3983 (rank 1 conflict city) to 141 (rank 40), which yields percentage changes of 3% and 14–19% to the distribution parameter. This shows that the distribution given for the attack intensity and frequency is robust across different urban scales and climates for large samples of data (see K–L divergence in figure 2). Figure 1$i$–$k$ shows three example cities and their diverse population size (varies by one order of magnitude), diverse city area (varies by one order of magnitude) and different climates. A large comparison set of random cities in the top 40 conflict cities is given in figure 3 which shows a common statistical distribution across all of them.

## 2.3. Prediction accuracy and information loss

Exponential distributions are commonly associated with waiting time between random and memoryless events (i.e. Poisson point processes). Therefore, the fitted negative exponential distributions in figure 1 indicate that sequential attacks in each city are unrelated. A possible reason is that each terrorist attack depends on a large number of variables (i.e. organization, logistics, finance, personnel, evading detection and opportunity), which suppresses any dependency between attacks. One other interesting property means that the probability of the waiting time for the next attack is constant, irrespective of how much time has surpassed. That is to say, one only needs to understand one single parameter $\hat{\mu}_T$ in order to predict the time for the next attack, irrespective of when the last attack was. This is similarly true for the death-toll per attack: the number of deaths in the next attack is independent of the previous attack event's death-toll.

As an example, figure 4 illustrates a system for prediction of the next terrorist attack and the uncertainty of the terrorism process. The time to next attack $t_i$ is independent of the time since previous attack $t_{i-1}$, and in general the time $t$ between attacks is negative exponential distributed with dependency on the mean number of attacks $t \sim \exp(-A/T)$ and the death $d$ is power law with

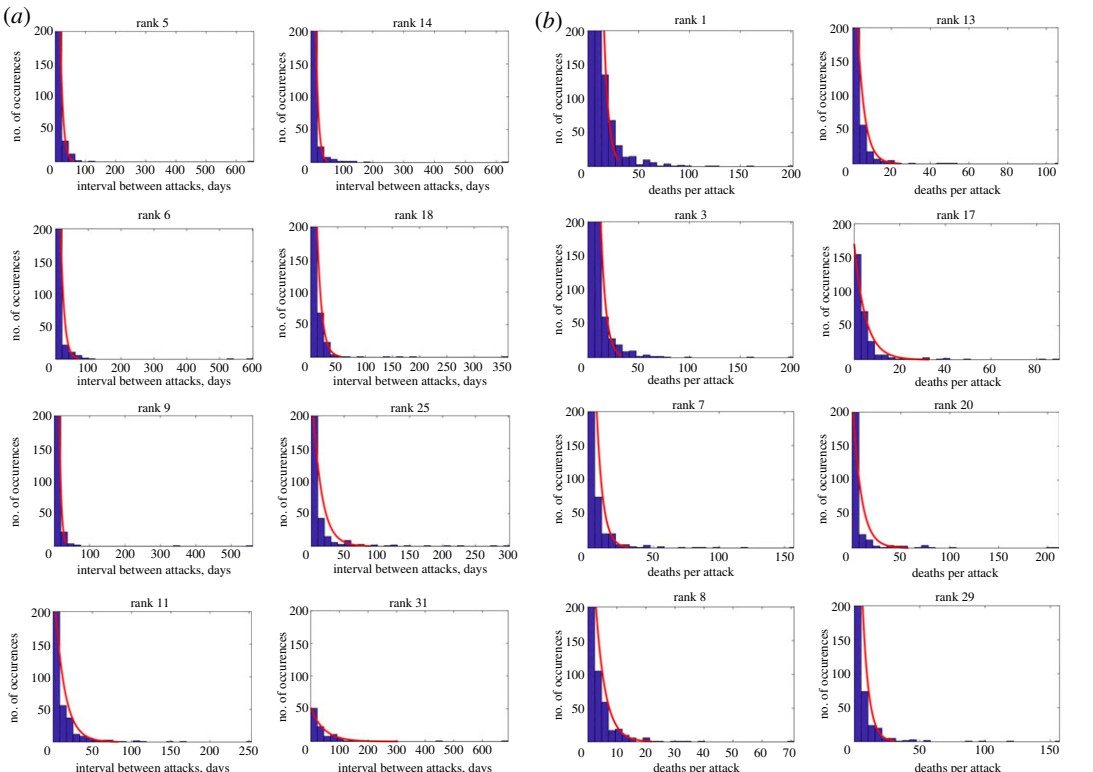

**Figure 3.** Terrorist attack interval and intensity in random sample of top 40 conflict cities with diverse urban scales and climates. (*a*) Common exponential distribution for attack intervals across random cities. (*b*) Common distribution for death-toll per attack across random cities.

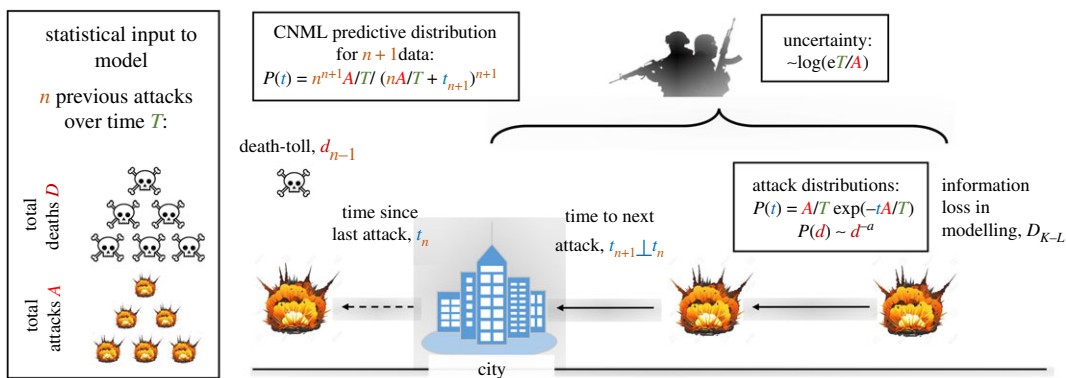

**Figure 4.** Prediction of next terrorist attack in city: the time to next attack $t_i$ is independent of the time since previous attack $t_{i-1}$, and in general the time between attacks is negative exponential distributed with dependency on the mean number of attacks $\sim$exp $(- A/T)$. The death-toll per attack (intensity) is power-law distributed. The uncertainty of the underlying terrorism process is logarithmically proportional to the mean time between attacks $\sim$log $(eT/A)$. The only input parameters into the model are the number of attacks $A$ and deaths $D$ aggregated over a period of $T$. CNML, conditional normalized maximum-likelihood.

dependency on the exponent $\alpha$. The uncertainty of the underlying terrorism process is logarithmically proportional to the mean time between attacks log $(eT/A)$. The only input parameters into the model are the number of attacks $A$ and deaths $D$ aggregated over a period of $T$. Among all continuous probability distributions with support $[0, +\infty)$ and mean $\hat{\mu}$, the exponential distribution has the largest entropy of log $(e\hat{\mu})$

$$h(X) = - \int_0^{+\infty} \frac{1}{\hat{\mu}} \exp\left(-\frac{x}{\hat{\mu}}\right) \log\left[\frac{1}{\hat{\mu}} \exp\left(-\frac{x}{\hat{\mu}}\right)\right] dx$$

$$= 1 - \log\left(\frac{1}{\hat{\mu}}\right) = \log(e\hat{\mu}).$$

(2.3)

This indicates a logarithmic higher information content (uncertainty) in the underlying terrorism processes (i.e. the terrorist organizations) that plan attacks with high waiting duration $\hat{\mu}_T$.

In prediction, we assume that for an observed city that has suffered $n$ previous attacks and these attacks have a frequency and intensity that are both exponentially distributed with unknown parameters. A common predictor for random and negative exponentially distributed data $x$ is to use is the ML predictor, which yields the following predictive density: $p_{\mathrm{ML}}(x_{n+1}\,|\,x_1, ..., x_n) = 1/\mu\,\exp\,(-(x_{n+1})/\mu)$, where $x_{n+1}$ is the future data value. An improved predictive distribution free of the issues of choosing priors is the conditional normalized maximum-likelihood (CNML) estimator, which yields the following predictive density of the future data [46]:

$$p_{\mathrm{CNML}}(x_{n+1}|x_1, ..., x_n) = \frac{n^{n+1}\mu^n}{(n\mu + x_{n+1})^{n+1}}, \qquad (2.4)$$

where $\mu$ is taken from the data ($\{x_1, ..., x_n\}$).

Given that both the time interval and the intensity (death-toll) fit negative exponential distributions, based on equation (4.1), the directed Kullback–Leibler (K–L) divergence (information loss) of adopting an exponential model instead of using the data is

$$D_{\mathrm{K-L}}(\mu\|\hat{\mu}) = \log\,(\mu^{-1}) - \log\,(\hat{\mu}^{-1}) + \frac{\mu}{\hat{\mu}} - 1, \qquad (2.5)$$

where $\mu$ is the mean of the data and $\hat{\mu}$ is the parameter of the regression or predictor. In general, as shown in figure 2, the negative exponential model causes a loss in information that varies between 1.2 and 2.8 nats (0.7–3.3 bits). This demonstrates that despite the seemingly accurate statistical characterization of conflict frequency, there is a non-negligible surprise element from an information-theoretic sense. Compared to other accuracy measures such as root mean square error (RMSE), the K–L divergence tells us how much information is lost in creating a predictive model based on our assumption that it fits a certain exponential distribution.

## 2.4. Spectrogram analysis

The frequency and intensity of violence have shifted over the past decade and this will affect the long-term accuracy of the proposed prediction model. Spectral analysis has the potential to observe the different frequency components of attacks and how they shift with time (2002–2014). As a hypothesis, it can potentially distinguish low-frequency (long time interval $T/A$) high-casualty (death-toll $D/A$) attacks from high-frequency low-casualty attacks. Spectral analysis using short-time-Fourier transform (STFT) is used on each city to produce spectrogram plots. The parameters used are Hamming window of size 128 (days), with non-overlap size of 120, 128 fast-Fourier transform sampling points to calculate the discrete Fourier transform. In general, the STFT of a discrete sequence $x[n]$ is defined as: $X(m, \omega) = \sum_{-\infty}^{+\infty} x[n]w[n - m]\exp\,(-j\omega n)$, where $w[n]$ is the window function and $\omega$ is the continuous frequency, and the spectrogram is defined as $|X(m, \omega)|^2$.

In figure 5, the results show the magnitude of the frequency of attacks as a function of time. Figure 5a-left shows the growing number of attacks and figure 5a-right shows that the deaths per attack is uncorrelated with the attack frequency. Digging deeper using spectrogram analysis, the author shows that there is a rapid growth ($G = 1.2–0.5$) in the magnitude of low-frequency attacks (b), and a slow growth ($G = 0.3–0.2$) in the magnitude of high-frequency attacks (c). The results indicate that the growth in the number of attacks and deaths is largely due to low-frequency attacks, which are causing a disproportionately high number of deaths (see the high variance in (a-right)). Figure 5d,e shows the spectrogram of the attacks for two example cities, with slices of low- and high-frequency magnitude variations as a function of time (days). The growing number of attacks (and deaths) is not due to population increases, as shown by the lack of correlation in figure 6.

The spectrogram trend we see is not universal, but the insight is general. That is to say, an abundance of small-casualty attacks gives our prediction greater accuracy (less information loss), whereas fewer high-casualty attacks leads to a poor estimation of the statistical parameter(s) used for prediction—leading to growing uncertainty in predictions. Specifically, we see greater high-casualty attacks in Iraq and Afghanistan, but not so in Colombia and Mexico.

In summary, the spectrogram analysis reveals that the growth in death-tolls from terrorist attacks seems to be due to an increase in the number of slower (1 attack per 100 days) and bigger-casualty attacks (over 10 deaths per attack). Referring back to the entropy of the terrorism process (i.e. entropy is $\log\,(eT/A)$), the growing number of low-frequency (large $T/A$) and high death-toll indicates that

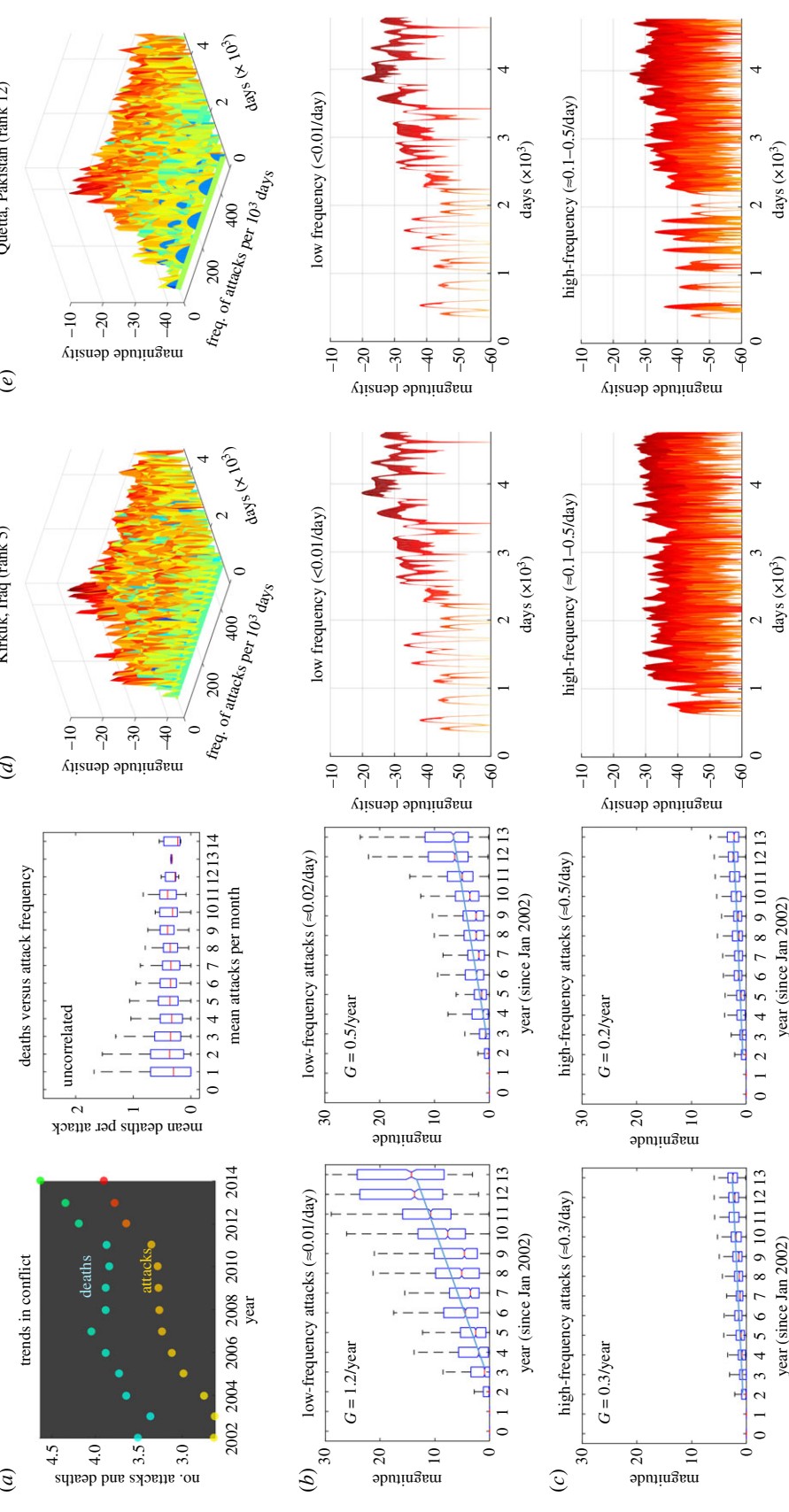

**Figure 5.** Temporal variations in the frequency of attacks: (*a*) the growing number of attacks and deaths and the death per attack is uncorrelated with the attack frequency. (*b*) The rapid growth ($G = 1.2$–$0.5$) in the magnitude of low-frequency attacks. (*c*) The slow growth ($G = 0.3$–$0.2$) in the magnitude of high-frequency attacks. (*d,e*) The spectrogram of the attacks between 2012 and 2014 for two example cities, with slices of low- and high-frequency magnitude variations as a function of time (days).

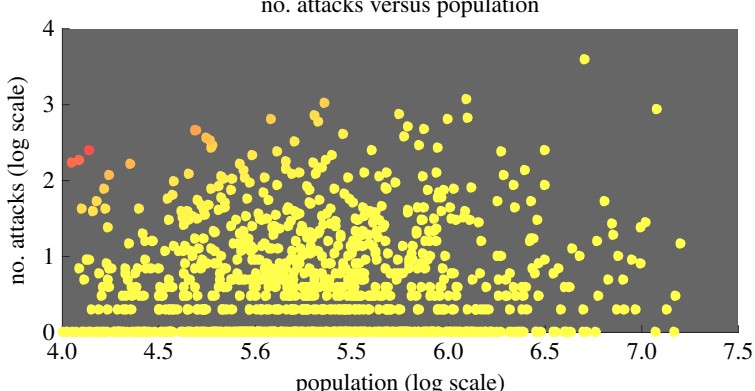

**Figure 6.** Attacks are uncorrelated with the population of the city.

the underlying terrorism process and organization is increasing in uncertainty. It is unclear how to combine the entropy measures if the high death-toll attacks are dependent on the longer planning process, and this is the focus of future research.

## 3. Discussions and conclusion

Conflict has transformed over recent human history. Post-Cold War conflict is dominated by political violence, interleaved with serious international criminal activities and ethnolinguistic civil war [47]. Post 9/11 conflict on the Eurasian continent is also dominated by counter-terrorism and Islamic violence [48,49]. Data collection on understanding these different facets of conflict has been critical to quantitative research in the political and social sciences [50].

Despite the seemingly complex reasons that drive modern terrorism, conflict and violence, this paper has shown that all modern conflicts exhibit common frequency and intensity patterns that can be modelled accurately to give true predictive powers to smart city systems. By examining over 30 000 geo-tagged conflict data points over 13 recent years, the paper demonstrates the following. The number of attacks and the death-toll is uncorrelated to the population of the city. The attacks are separated by a time that is negative exponentially distributed $\sim \exp(-\hat{\mu}_T)$ and the number of deaths per attack follows power-law distributions. The prediction parameters explain 69–87% of the variations in real data. While the parameters of the distributions vary between cities and with time, these findings show that the frequency of terror attacks is random and memoryless. By memoryless, we mean that the probability of a new attack is largely or entirely independent of previous attacks. This seems contradictory to some existing studies which suggest excitation and de-escalation effects [18,20]. This could be because across a city, we have averaged out micro-scale dynamics; but it is nonetheless thought provoking from a prediction point of view. That is to say, understanding historical attacks (and its pattern) gives no extra predictive power to future attacks at the city scale, and is true for all cities suffering all genres of conflict (e.g. from Mosul to Medellin).

The distributions found in this paper can be used to predict the next attack and the K–L divergence is used to show that approximately 0.7–3.3 bits of information is lost through the predictions. As such, future work should focus on integrating generalized statistical models presented in this paper with microscopic excitation and mechanical models [20], as well as detailed causal mechanisms related to resources [51] and climate [52].

Using spectrogram analysis, it was uncovered that the growth in death-tolls is due to a growing number of slower but higher-casualty attacks. Combining the spectrogram analysis with the entropy analysis, the combined results seem to indicate a logarithmically increasing uncertainty in the underlying terrorism random process. The growing uncertainty could be because the causal mechanisms are becoming more multifaceted and cannot be pinpointed to a well-defined set of mechanisms (e.g. inequality and grievances [53–55]). This is, of course, speculative at this point, but this growing uncertainty makes prediction and developing counter-terrorism strategies more challenging.

**Table 1.** Top 40 prominent conflict cities.

| city (1–20) | country | city (21–40) | country |
|---|---|---|---|
| Baghdad | Iraq | Musayyib | Iraq |
| Mosul | Iraq | Banghazi | Libya |
| Baqubah | Iraq | Farah | Afghanistan |
| Karachi | Pakistan | Zareh | Afghanistan |
| Kirkuk | Iraq | Jalalabad | Afghanistan |
| Lashkar | Afghanistan | Kabul | Afghanistan |
| Peshawar | Pakistan | Ghazni | Afghanistan |
| Mogadishu | Somalia | Saidu | Pakistan |
| Yala | Thailand | Kohat | Pakistan |
| Fallujah | Iraq | Pattani | Thailand |
| Kandahar | Afghanistan | Qalat | Afghanistan |
| Quetta | Pakistan | Cotabato | Philippines |
| Asadabad | Afghanistan | Tall Afar | Iraq |
| Tikrit | Iraq | Meymaneh | Afghanistan |
| Ramadi | Iraq | Qasr Shirin | Iran |
| Bannu | Pakistan | Gardiz | Afghanistan |
| Parachinar | Pakistan | Baraki | Afghanistan |
| Narathiwat | Thailand | Sukkur | Pakistan |
| Samarra | Iraq | Groznyy | Russia |
| Maiduguri | Nigeria | Tizi-Ouzou | Algeria |

# 4. Methodology

## 4.1. Data

The terrorism and conflict data used in this paper is sourced from 30 000+ attacks between 2002 and 2014 (13 years) from the GTD [41]. For each city and over time period $T$, the GTD contains the number of geo-tagged attacks $A$ and death-toll $D$ from incidents, which range from small-scale assassinations (one death) to large-scale massacres (thousands dead). A plot of the major terrorist and conflict incidents is shown in figure 1$a$. The GTD data are then clustered to the nearest city.[1] As a result of clustering, the author shows that the vast majority of conflict incidents occur in close proximity to an urban area with a mean clustering distance of 27 km. This means while most data points are in cities, some do occur in rural and suburban areas, which is still relevant from a policy perspective. It is worth noting that the number of attacks (and deaths) is not due to population increases, as shown by the lack of correlation in figure 6. Therefore, models with predictive power are needed to understand the frequency and intensity of attacks for each city.

Using the data, two variables are extracted: (i) the time interval $t$ between each consecutive attack (frequency), and (ii) the death-toll per attack $d$ (intensity). Of the data obtained between 2002 and 2014, only 40 cities in the world have sufficient conflict data to obtain distributions from which the error in ML parameter estimation is less than 20% (see Results section). These cities range from the Middle East, West Africa, South Asia, to the Far East. A list of the cities can be found in table 1.

## 4.2. Metrics

In order to compare datasets, the coefficient of determination $R^2$ is used. It is a number that indicates how well the statistical regression model fits the data. In other words, the percentage of variance in

---

[1]Over 7000 cities and settlements were considered sourced from the National Geospatial Intelligence Agency [56].

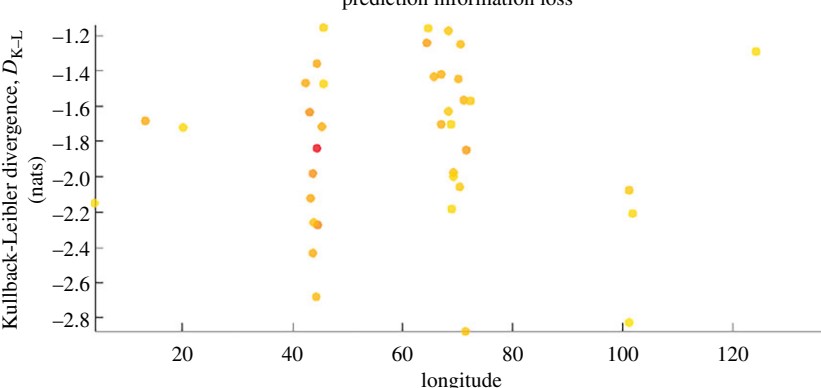

**Figure 7.** Accuracy and information loss in prediction: directed K–L divergence (information loss) for predicting the time interval between attacks for cities in different locations. Colour indicates the number of casualties (red is highest).

the data that can be explained by the proposed model. For a data vector $y = [y_1, y_2, \ldots y_K]$ (with mean $\overline{y}$) and a predicted data vector using the regression model $\hat{y}$, the residue vector is defined as $e = y - \hat{y}$. The coefficient of determination $R^2$ is defined as $R^2 \equiv 1 - \sum_k e_k^2 / \sum_k (y_k - \overline{y})^2$, where the numerator is known as the residual sum of squares and the denominator is known as the total sum of squares. In this paper, the analysis employs the adjusted $R^2$ to take discount against extra variables in the model adjusted $R^2 = 1 - (1 - R^2)((K - 1)/(K - V - 1))$, where $V$ is the number of variables in the regression model.

In order to compare between different probability distributions $P$ (true data) and $Q$ (regression model), the directed information gain/loss metric is used. The K–L divergence of $Q$ from $P$ is $D_{K-L}(P\|Q)$, and it is defined as [57]

$$D_{K-L}(P\|Q) = \int_{-\infty}^{+\infty} p(x) \log \frac{p(x)}{q(x)} \, dx, \tag{4.1}$$

where $p(x)$ and $q(x)$ denote the densities of $P$ and $Q$. For exponential distributions, this is given as: $D_{K-L}(\mu\|\hat{\mu}) = \log(\mu^{-1}) - \log(\hat{\mu}^{-1}) + \mu/\hat{\mu} - 1$. Robustness of the results shown in figure 7 across cities of different longitudes demonstrates that the results are valid for different regions of the world.

Ethics. No animal or human testing is involved in this study.

Data accessibility. Terrorism data is available from GTD database [41], and cities data are available within the Dryad Digital Repository: https://doi.org/10.5061/dryad.cj8kk41 [42]. All visualization maps are rendered from OpenStreetMap and all pictures are drawn by the author.

Authors' contributions. W.G. conceived the idea, sourced and analysed the data, and wrote the paper.

Competing interests. I declare I have no competing interests.

Funding. W.G. is partly funded by the Alan Turing Institute under the EPSRC grant no. EP/N510129/1.

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
