## [Reviewer comments · Royal Society Open Science]

Review History

RSOS-190645.R0 (Original submission)

Review form: Reviewer 1 (Akin Unver)

Is the manuscript scientifically sound in its present form?

Yes

Are the interpretations and conclusions justified by the results?

Yes

Is the language acceptable?

Yes

Do you have any ethical concerns with this paper?

No

Have you any concerns about statistical analyses in this paper?

No

Recommendation?

Accept with minor revision (please list in comments)

Comments to the Author(s)

Comments are listed in the attached review document (Appendix A).

Review form: Reviewer 2**Is the manuscript scientifically sound in its present form?**

Yes

Are the interpretations and conclusions justified by the results?

Yes

Is the language acceptable?

Yes

Do you have any ethical concerns with this paper?

No

Have you any concerns about statistical analyses in this paper?

Yes

Recommendation?

Major revision is needed (please make suggestions in comments)

Comments to the Author(s)

I have read the manuscript RSOS-190645 'Common Statistical Patterns in Urban Terrorism' with much interest and it presents important insights to the relationship between the interval and intensity of terrorist attacks. However, there are some points that, from my perspective, need to be addressed before publication.

1) Framing: The framing of the paper could highlight the main contribution more clearly. My reading of the paper is that the findings in regard to the interval between attacks are most central and that the intensity findings (Results a page 3) are more of a side note.

2) Literature: There is an early literature on the linkage between severity and duration of conflict that might be relevant for this this manuscript to consider. Weiss, 1993; Klingberg 1966; Voevodsky, 1969.

3) GTD database. There probably needs to be a short discussion about how terrorism is defined (especially because the scientific and political use of the term can differ) in the context of the GTD database and the potential overlap with events that one could also consider asymmetric warfare (especially events in Iraq and Afghanistan). Also a short note on why the whole GTD is used and not a particular subset (e.g. target type or attack type) would be helpful.

4) Page 6 last paragraph. The manuscript claims that 'the distribution given the attack intensity is robust across different urban scales and climates'. It is not clear how this robustness is established.

5) Page 7 first paragraph. The paragraph states that sequential attacks in each city are unrelated and that a 'possible reason is that each terrorist attack depends on a large number of variables (i.e., organization, logistics, finance, personal, evading detection, and opportunity). This raises

question about the general modeling approach. Couldn't the relationship between number of attacks and time interval be modeled controlling for a number of factors?

6) The motivation for using the Kullback-Leibler Divergence to assess model fit/performance is not clear. How is this better than just comparing expected time intervals to true intervals (e.g. RMSE). In addition, it was not clear whether the predictions are assessed within the training sample or whether they were assessed on a test sample. It would be recommendable to assess predictions on a test-sample.

7) The motivation to use the spectrogram analysis could be made clearer on page 3 sixth paragraph. The main finding is that the growth in death-tolls from terrorist attacks is 'due to an increase in the number of slower and bigger casualty attacks'. It would be good to explore or at least discuss to which extent this might be due to coding bias in GTD (see Iraq/Afghanistan and temporal trends in the data.).

Decision letter (RSOS-190645.R0)

09-Jul-2019

Dear Dr Guo,

The editors assigned to your paper ("Common Statistical Patterns in Urban Terrorism") have now received comments from reviewers. We would like you to revise your paper in accordance with the referee and Associate Editor suggestions which can be found below (not including confidential reports to the Editor). Please note this decision does not guarantee eventual acceptance.

Please submit a copy of your revised paper before 01-Aug-2019. Please note that the revision deadline will expire at 00.00am on this date. If we do not hear from you within this time then it will be assumed that the paper has been withdrawn. In exceptional circumstances, extensions may be possible if agreed with the Editorial Office in advance. We do not allow multiple rounds of revision so we urge you to make every effort to fully address all of the comments at this stage. If deemed necessary by the Editors, your manuscript will be sent back to one or more of the original reviewers for assessment. If the original reviewers are not available, we may invite new reviewers.

- Data accessibility

<http://datadryad.org/submit?journalID=RSOS&manu=RSOS-190645>

- Competing interests

- Authors' contributions

- Acknowledgements

- Funding statement

on behalf of Dr Hamed Haddadi (Associate Editor) and Marta Kwiatkowska (Subject Editor)
openscience@royalsociety.org

Associate Editor's comments (Dr Hamed Haddadi):

Many thanks for your submission to RSOS. Please visit the reviewers' comments and prepare a revision accordingly.

Comments to Author:

Reviewers' Comments to Author:
Reviewer: 1

Comments to the Author(s)
Comments are listed in the attached review document.

Reviewer: 2

Comments to the Author(s)

I have read the manuscript RSOS-190645 'Common Statistical Patterns in Urban Terrorism' with much interest and it presents important insights to the relationship between the interval and intensity of terrorist attacks. However, there are some points that, from my perspective, need to be addressed before publication.

1) Framing: The framing of the paper could highlight the main contribution more clearly. My reading of the paper is that the findings in regard to the interval between attacks are most central and that the intensity findings (Results a page 3) are more of a side note.

2) Literature: There is an early literature on the linkage between severity and duration of conflict that might be relevant for this this manuscript to consider. Weiss, 1993; Klingberg 1966; Voevodsky, 1969.

3) GTD database. There probably needs to be a short discussion about how terrorism is defined (especially because the scientific and political use of the term can differ) in the context of the GTD database and the potential overlap with events that one could also consider asymmetric warfare (especially events in Iraq and Afghanistan). Also a short note on why the whole GTD is used and not a particular subset (e.g. target type or attack type) would be helpful.

4) Page 6 last paragraph. The manuscript claims that 'the distribution given the attack intensity is

robust across different urban scales and climates'. It is not clear how this robustness is established.

5) Page 7 first paragraph. The paragraph states that sequential attacks in each city are unrelated and that a `possible reason is that each terrorist attack depends on a large number of variables (i.e., organization, logistics, finance, personal, evading detection, and opportunity). This raises question about the general modeling approach. Couldn't the relationship between number of attacks and time interval be modeled controlling for a number of factors?

6) The motivation for using the Kullback-Leibler Divergence to assess model fit/performance is not clear. How is this better than just comparing expected time intervals to true intervals (e.g. RMSE). In addition, it was not clear whether the predictions are assessed within the training sample or whether they were assessed on a test sample. It would be recommendable to assess predictions on a test-sample.

7) The motivation to use the spectrogram analysis could be made clearer on page 3 sixth paragraph. The main finding is that the growth in death-tolls from terrorist attacks is `due to an increase in the number of slower and bigger casualty attacks'. It would be good to explore or at least discuss to which extent this might be due to coding bias in GTD (see Iraq/Afghanistan and temporal trends in the data.).

Author's Response to Decision Letter for (RSOS-190645.R0)

See Appendix B.

RSOS-190645.R1 (Revision)

Review form: Reviewer 1 (Akin Unver)

Is the manuscript scientifically sound in its present form?

Yes

Are the interpretations and conclusions justified by the results?

Yes

Is the language acceptable?

Yes

Do you have any ethical concerns with this paper?

No

Have you any concerns about statistical analyses in this paper?

No

Recommendation?

Accept as is

Comments to the Author(s)

The revisions sufficiently address my fundamental concerns about engaging with the existing scholarship on terrorism. The theoretical thrust of the paper is now much clearer and can offer citeable findings for the relevant sub-disciplines in terrorism research. Furthermore, the authors have clarified their findings on the memoryless aspect of terrorism and report to what degree spatial and temporal dynamics can and cannot explain the occurrence of terrorist attacks. In addition, these authors are now well-defended against possible attacks from the social science terrorism research fields on 'reinventing the wheel', because they address the most core works in this field.

I find this study novel in its methods and empirical testing of existing theories of terrorism and believe that the study, in its current form, is a rigorous example of how computer science can contribute to our understanding of violence and how social scientists can use computational tools to harness violence event data.

I recommend acceptance in its current form.

Review form: Reviewer 2

Is the manuscript scientifically sound in its present form?

Yes

Are the interpretations and conclusions justified by the results?

Yes

Is the language acceptable?

Yes

Do you have any ethical concerns with this paper?

No

Have you any concerns about statistical analyses in this paper?

No

Recommendation?

Accept as is

Comments to the Author(s)

Thank you for addressing my concerns effectively.

Decision letter (RSOS-190645.R1)

16-Aug-2019

Dear Dr Guo,

I am pleased to inform you that your manuscript entitled "Common Statistical Patterns in Urban Terrorism" is now accepted for publication in Royal Society Open Science.

on behalf of Prof Marta Kwiatkowska (Subject Editor)
openscience@royalsociety.org

Associate Editor Comments to Author:

The reviewers have assessed this revision and now recommend publication - congratulations and thank you for submitting to Royal Society Open Science!

Reviewer comments to Author:

Reviewer: 2

Comments to the Author(s)

Thank you for addressing my concerns effectively.

Reviewer: 1

Comments to the Author(s)

The revisions sufficiently address my fundamental concerns about engaging with the existing scholarship on terrorism. The theoretical thrust of the paper is now much clearer and can offer citeable findings for the relevant sub-disciplines in terrorism research. Furthermore, the authors have clarified their findings on the memoryless aspect of terrorism and report to what degree spatial and temporal dynamics can and cannot explain the occurrence of terrorist attacks. In addition, these authors are now well-defended against possible attacks from the social science terrorism research fields on 'reinventing the wheel', because they address the most core works in this field.

I find this study novel in its methods and empirical testing of existing theories of terrorism and believe that the study, in its current form, is a rigorous example of how computer science can contribute to our understanding of violence and how social scientists can use computational tools to harness violence event data.

I recommend acceptance in its current form.

Appendix A

This paper aims to tackle an important question: what drives spatial and temporal dynamics of terrorist attacks. In doing so, the study relies on Global Terrorism Database (GTD) through 2002-14, by focusing on 30,000 geo-tagged terrorism acts over 7000 cities worldwide. I'm happy to see that hard sciences scholars are developing interest in what is primarily and essentially a social sciences research question. The technical aspect of the paper is sound, but my comments are mostly directed towards how this paper can contribute to terrorism research by engaging more with the terrorism scholarship and by better contextualizing its results.

1. The paper will benefit a lot from engaging newer studies on the role of geography on terrorism to avoid re-inventing the wheel on how spatio-temporal dynamics affect conflict. Some good examples that both offer significant results and also provide excellent literature reviews for the author's benefit are:
 - Vogt, Manuel, Nils-Christian Bormann, Seraina Rüeegger, Lars-Erik Cederman, Philipp Hunziker, and Luc Girardin. "Integrating data on ethnicity, geography, and conflict: The ethnic power relations data set family." *Journal of Conflict Resolution* 59, no. 7 (2015): 1327-1342.
 - Buhaug, Halvard, Scott Gates, and Päivi Lujala. "Geography, rebel capability, and the duration of civil conflict." *Journal of Conflict Resolution* 53, no. 4 (2009): 544-569.
 - Weidmann, Nils B. "Geography as motivation and opportunity: Group concentration and ethnic conflict." *Journal of Conflict Resolution* 53, no. 4 (2009): 526-543.
 - Buhaug, Halvard, and Scott Gates. "The geography of civil war." *Journal of Peace Research* 39, no. 4 (2002): 417-433.
 - Flint, Colin. "Terrorism and counterterrorism: Geographic research questions and agendas." *The Professional Geographer* 55, no. 2 (2003): 161-169.
 - Toft, Monica Duffy. *The geography of ethnic violence: Identity, interests, and the indivisibility of territory*. Princeton University Press, 2005.
 - Enders, Walter, and Todd Sandler. "Distribution of transnational terrorism among countries by income class and geography after 9/11." *International Studies Quarterly* 50, no. 2 (2006): 367-393.
 - Findley, Michael G., and Joseph K. Young. "Terrorism and civil war: A spatial and temporal approach to a conceptual problem." *Perspectives on Politics* 10, no. 2 (2012): 285-305.
 - Krieger, Tim, and Daniel Meierrieks. "What causes terrorism?." *Public Choice* 147, no. 1-2 (2011): 3-27.
 - Gao, Peng, Diansheng Guo, Ke Liao, Jennifer J. Webb, and Susan L. Cutter. "Early detection of terrorism outbreaks using prospective space-time scan statistics." *The Professional Geographer* 65, no. 4 (2013): 676-691.
2. The theoretical thrust of the paper is that there are tangible spatial determinants of terrorism at the city-level, and that 'terror attacks are seemingly random and memoryless for all global cities.' And that 'here appears to be an increase in the uncertainty over the predictability of attacks, challenging our ability to develop effective counter-terrorism capabilities.' Both of these findings are already well-evidenced in the existing terrorism literature. The paper has to make a more convincing case to argue that the spatial analysis conducted here goes beyond our existing limitations on predicting terrorist attacks in the cities. Mainly, by engaging with the scholarship above (and other relevant ones they cite), the study has to offer a truly new finding so that terrorism researchers can benefit and most importantly, cite.
3. The paper reports '*a growing uncertainty hidden in the complex process*'. However, if the study will report the same uncertainty that other past studies did, it has to answer what

additional knowledge it has created to add to the theoretical literature. The easiest critique to this paper is the fact that it completely omits two of the actual main drivers of terrorism: inequality and grievances (Collier, Paul, Anke Hoeffler, and Dominic Rohner. "Beyond greed and grievance: feasibility and civil war." *oxford Economic papers* 61, no. 1 (2009): 1-27. Ballentine, Karen, ed. *The political economy of armed conflict: Beyond greed and grievance*. Lynne Rienner Publishers, 2003., Berdal, Mats R., Mats Berdal, and David Malone, eds. *Greed & grievance: Economic agendas in civil wars*. Lynne Rienner Publishers, 2000.). Without controlling for these two core dynamics, simply using urban ecology as an independent variable will not yield a strong theoretical finding.

4. The terrorism studies field already knows that terrorists conduct terrorist attacks in urban areas, both for casualty maximization, and also for media and attention exposure. The study could offer a more convincing sample selection rationale (i.e. why cities) in line with the 'terrorist target selection' literature: (Asal, Victor H., R. Karl Rethemeyer, Ian Anderson, Allyson Stein, Jeffrey Rizzo, and Matthew Rozea. "The softest of targets: A study on terrorist target selection." *Journal of Applied Security Research* 4, no. 3 (2009): 258-278. Drake, Charles JM. "The role of ideology in terrorists' target selection." *Terrorism and Political Violence* 10, no. 2 (1998): 53-85. Drake, Charles JM, David Drake, and Freud. *Terrorists' target selection*. London: Macmillan, 1998.)
5. Looking at post-9/11 (2002-14) terrorism patterns alone will provide us with overwhelming Islamist terrorism data. In contrast, prior to the Cold War, ideological drivers of terrorism used to be more nuanced. The paper must acknowledge that looking at post-9/11 will not provide us with a 'general theory of terrorism', but a temporally specific and contextual theory on Islamic terrorism alone. This is fine, because data became more available as US government agencies began funding dedicated research and data initiatives (like GTD) in the aftermath of 9-11, but of course, this temporal selection can't be generalized. The study has to clarify this critical point by perhaps changing the title and the abstract to let readers know that it is engaging overwhelmingly with Islamist terrorism. This is a very crucial distinction, because the study doesn't use the data on IRA, ETA and other European-American terrorist groups active through the Cold War.
6. 'The interesting observation is that most previous studies have considered low resolution conflicts (major wars) that span over 100 years, and it seems that the power law distribution remains valid even for high resolution terrorism and non-conventional conflict data in the modern era.' - This sentence is unclear. Does the author mean 'low-resolution conflict' to imply less granular data at the national-level? If so, sub-national data are in fact pretty much on the rise in terrorism studies; most newer studies combine national-level with sub-national (regional) data. This is not a valid criticism of the studies in the field in the last several years. Please check: (Urdal, Henrik. "Population, resources, and political violence: A subnational study of India, 1956-2002." *Journal of Conflict Resolution* 52, no. 4 (2008): 590-617. Maume, Michael O., and Matthew R. Lee. "Social institutions and violence: A sub-national test of institutional anomie theory." *Criminology* 41, no. 4 (2003): 1137-1172. Raleigh, Clionadh. "Violence against civilians: A disaggregated analysis." *International Interactions* 38, no. 4 (2012): 462-481. Homer-Dixon, Thomas F. "Environmental scarcities and violent conflict: evidence from cases." *International security* 19, no. 1 (1994): 5-40. Witmer, Frank DW, Andrew M. Linke, John O'Loughlin, Andrew Gettelman, and Arlene Laing. "Subnational violent conflict forecasts for sub-Saharan Africa, 2015-65, using climate-sensitive models." *Journal of Peace Research* 54, no. 2 (2017): 175-192.)
7. When and where terrorists attack are very context-specific behavioral types. Terrorist behavior in Iraq diverges from those in Syria, versus Ukraine, and others do so because such behavior is non-random. These are driven by intra-organizational dynamics (leadership challenge, preventing splits and bolster group identity), external resilience

(capacity of state security agencies, terrain, social cohesion and social support for terrorist groups) and ideology of the group (religious, ethno-nationalist etc.) Where and how frequent ISIS attacks in Iraq is different than its behavior in Syria. There are even further divergences between terrorism in urban areas of failed (failing) states, democracies and authoritarian countries. Trying to impose a general 'supra-level' explanation to these varieties will yield inconclusive results, and the study reports this inconclusivity itself. The study can escape this deadlock by cross-checking with the UCDP/PRIO data (or ACLED) and see whether non-terrorist sub-national violence has any direct correlation with terrorism. This will enable the author to distinguish between areas that are suffering BOTH from high intensity of conflict AND terrorism, and those where the two phenomena are not directly related (high violence, low terrorism or low violence, high terrorism).

8. I think the methodical sophistication of the paper should more directly and robustly interact with the terrorism studies literature, especially with those studies that explore how spatio-temporal dynamics interact with the more relevant and important HUMAN and SOCIAL drivers of terrorism.

While I understand the author's models, I'm a social scientist and the models themselves must be reviewed by a reviewer from mathematics, statistics or econometrics fields. But revising the paper in line of above comments will make this a highly citable and useful study for the terrorism literature.

Appendix B

Response to Reviewer Comments

Reviewer: 1 (Copied from Attachment)

General Comment. This paper aims to tackle an important question: what drives spatial and temporal dynamics of terrorist attacks. In doing so, the study relies on Global Terrorism Database (GTD) through 2002-14, by focusing on 30,000 geo-tagged terrorism acts over 7000 cities worldwide.

I'm happy to see that hard sciences scholars are developing interest in what is primarily and essentially a social sciences research question. The technical aspect of the paper is sound, but my comments are mostly directed towards how this paper can contribute to terrorism research by engaging more with the terrorism scholarship and by better contextualizing its results.

General Response. Dear reviewer, thank you for taking the interest and time to read my manuscript. I have gone through your comments and recommended reading and tried to address the comments. In most cases, I believe I have addressed them, but there are 1-2 cases where there is absence of data at the city level for the confounding variables you speak of, and I can only offer a speculative discussion and point towards further research. In both this document and the revised manuscript, I have highlighted revised new or significantly changed text in blue.

Comment 1. The paper will benefit a lot from engaging newer studies on the role of geography on terrorism to avoid re-inventing the wheel on how spatio-temporal dynamics affect conflict. Some good examples that both offer significant results and also provide excellent literature reviews for the author's benefit are:

- Vogt, Manuel, Nils-Christian Bormann, Seraina Rügger, Lars-Erik Cederman, Philipp Hunziker, and Luc Girardin. "Integrating data on ethnicity, geography, and conflict: The ethnic power relations data set family." *Journal of Conflict Resolution* 59, no. 7 (2015): 1327-1342.
- Buhaug, Halvard, Scott Gates, and Päivi Lujala. "Geography, rebel capability, and the duration of civil conflict." *Journal of Conflict Resolution* 53, no. 4 (2009): 544-569.
- Weidmann, Nils B. "Geography as motivation and opportunity: Group concentration and ethnic conflict." *Journal of Conflict Resolution* 53, no. 4 (2009): 526-543.
- Buhaug, Halvard, and Scott Gates. "The geography of civil war." *Journal of Peace Research* 39, no. 4 (2002): 417-433.
- Flint, Colin. "Terrorism and counterterrorism: Geographic research questions and agendas." *The Professional Geographer* 55, no. 2 (2003): 161-169.
- Toft, Monica Duffy. *The geography of ethnic violence: Identity, interests, and the indivisibility of territory.* Princeton University Press, 2005.
- Enders, Walter, and Todd Sandler. "Distribution of transnational terrorism among countries by income class and geography after 9/11." *International Studies Quarterly* 50, no. 2 (2006): 367-393.
- Findley, Michael G., and Joseph K. Young. "Terrorism and civil war: A spatial and temporal approach to a conceptual problem." *Perspectives on Politics* 10, no.2 (2012): 285-305.

- Krieger, Tim, and Daniel Meierrieks. "What causes terrorism?." Public Choice 147, no. 1-2 (2011): 3-27.
- Gao, Peng, Diansheng Guo, Ke Liao, Jennifer J. Webb, and Susan L. Cutter. "Early detection of terrorism outbreaks using prospective space–time scan statistics." The Professional Geographer 65, no. 4 (2013): 676-691.

Response 1. The above references are very useful, and we have interleaved their findings and re-written our findings in their context. Specifically, we have added two paragraphs to discuss the aforementioned findings:

Introduction: *Detailed causal mechanisms on why certain locations experience more conflict or longer duration conflict have been studied \cite{Krieger11}. For example, regions far from government control and rich in natural resources tend to experience protracted conflict \cite{Halvard02, Halvard09, Toft05}. However, not all conflict is driven by such mechanisms, e.g. protracted urban warfare in Colombian cities. It has also been shown that it is difficult to separate the different genres of conflict, e.g. civil war and terrorism \cite{Findley12} and as such it does make sense to consider them together from a statistical modeling or prediction perspective.*

Discussion: *Conflict has transformed over recent human history. Post Cold War conflict is dominated by political violence, interleaved with serious international criminal activities and ethnolinguistic civil war \cite{Ugarriza09}. Post 9/11 conflict on the Eurasian continent is also dominated by counter-terrorism and Islamic violence \cite{Flint03, Enders06}. Data collection on understanding these different facets of conflict has been critical to quantitative research in the political and social sciences \cite{Vogt15}.*

Comment 2. The theoretical thrust of the paper is that there are tangible spatial determinants of terrorism at the city-level, and that ‘terror attacks are seemingly random and memoryless for all global cities.’ And that ‘here appears to be an increase in the uncertainty over the predictability of attacks, challenging our ability to develop effective counter-terrorism capabilities.’ Both of these findings are already well-evidenced in the existing terrorism literature. The paper has to make a more convincing case to argue that the spatial analysis conducted here goes beyond our existing limitations on predicting terrorist attacks in the cities. Mainly, by engaging with the scholarship above (and other relevant ones they cite), the study has to offer a truly new finding so that terrorism researchers can benefit and most importantly, cite.

Response 2. Dear reviewer, having read the above papers, I don’t think the memoryless property has been shown for individual cities. By memoryless, we mean that the probability of a new attack is largely or entirely independent of previous attacks. This seems contradictory to some existing studies which suggest excitation and de-escalation effects. This could be because across a city, we have averaged out micro-scale dynamics; but it is nonetheless thought provoking from a prediction point of view. That is to say, understanding historical attacks (and its pattern) gives no extra predictive power to future attacks at the city scale, and is true for all cities suffering all genres of conflict (e.g. from Mosul to Medellin). We have revised the paper to highlight this finding.

Comment 3. The paper reports 'a growing uncertainty hidden in the complex process'. However, if the study will report the same uncertainty that other past studies did, it has to answer what additional knowledge it has created to add to the theoretical literature. The easiest critique to this paper is the fact that it completely omits two of the actual main drivers of terrorism: inequality and grievances

Collier, Paul, Anke Hoeffler, and Dominic Rohner. "Beyond greed and grievance: feasibility and civil war." oxford Economic papers 61, no. 1 (2009): 1-27.

Ballentine, Karen, ed. The political economy of armed conflict: Beyond greed and grievance. Lynne Rienner Publishers, 2003.,

Berdal, Mats R., Mats Berdal, and David Malone, eds. Greed & grievance: Economic agendas in civil wars. Lynne Rienner Publishers, 2000.

Without controlling for these two core dynamics, simply using urban ecology as an independent variable will not yield a strong theoretical finding.

Response 3. Dear reviewer, our paper uses data to show that from a statistical perspective, conflict is transforming in a way that is making prediction less certain. That is to say, I can no longer predict as accurately as I did last year, and this is consistently true across the world. In addressing your suggestion, this could be because the causal mechanisms are becoming more multi-faceted and cannot be pinpointed to a well-defined set of mechanisms. This is of course speculative at this point, but you are right to suggest that we relate our findings back to the causal mechanisms. We have added a discussion at the end of the paper to this effect and included the above references as examples of key mechanisms that drive conflict.

Comment 4. The terrorism studies field already knows that terrorists conduct terrorist attacks in urban areas, both for casualty maximization, and also for media and attention exposure. The study could offer a more convincing sample selection rationale (i.e. why cities) in line with the 'terrorist target selection' literature:

Asal, Victor H., R. Karl Rethemeyer, Ian Anderson, Allyson Stein, Jeffrey Rizzo, and Matthew Rozea. "The softest of targets: A study on terrorist target selection." Journal of Applied Security Research 4, no. 3 (2009): 258-278.

Drake, Charles JM. "The role of ideology in terrorists' target selection." Terrorism and Political Violence 10, no. 2 (1998): 53-85.

Drake, Charles JM, David Drake, and Freud. Terrorists' target selection. London: Macmillan, 1998.

Response 4. Thank you, this is indeed useful from our justification perspective. We have added this and also reflected on the fact that rural attacks maybe under reported and need further investigation in the future.

Comment 5. Looking at post-9/11 (2002-14) terrorism patterns alone will provide us with overwhelming Islamist terrorism data. In contrast, prior to the Cold War, ideological drivers of terrorism used to be more nuanced. The paper must acknowledge that looking at post-9/11 will not provide us with a 'general theory of terrorism', but a temporally specific and contextual theory on Islamic terrorism alone.

This is fine, because data became more available as US government agencies began funding dedicated research and data initiatives (like GTD) in the aftermath of 9-11, but of course, this temporal selection can't be generalized. The study has to clarify this

critical point by perhaps changing the title and the abstract to let readers know that it is engaging overwhelmingly with Islamist terrorism. This is a very crucial distinction, because the study doesn't use the data on IRA, ETA and other European-American terrorist groups active through the Cold War.

Response 5. Thank you for this useful comment. As you pointed out, our main motivation for post 9/11 data analysis is the more consistent quality of geo-tagged event data across the whole world. As it is across the whole world, a significant portion of that data is not "War or Terror" related (e.g. Colombia, Narco-War Mexico, political violence in Thailand & India). We have clarified this point and hope the reviewer can agree (to an extent) that our analysis has some degree of generality to it, even if some of the top violence cities are in the war on terror area.

Comment 6. 'The interesting observation is that most previous studies have considered low resolution conflicts (major wars) that span over 100 years, and it seems that the power law distribution remains valid even for high resolution terrorism and non-conventional conflict data in the modern era.' – This sentence is unclear. Does the author mean 'low-resolution conflict' to imply less granular data at the national-level? If so, sub-national data are in fact pretty much on the rise in terrorism studies; most newer studies combine national-level with sub-national (regional) data. This is not a valid criticism of the studies. in the field in the last several years.

Please check:

Urdal, Henrik. "Population, resources, and political violence: A subnational study of India, 1956–2002." *Journal of Conflict Resolution* 52, no. 4 (2008): 590-617.

Maume, Michael O., and Matthew R. Lee. "Social institutions and violence: A sub-national test of institutional anomie theory." *Criminology* 41, no. 4 (2003): 1137-1172.

Raleigh, Clionadh. "Violence against civilians: A disaggregated analysis." *International Interactions* 38, no. 4 (2012): 462-481.

Homer-Dixon, Thomas F. "Environmental scarcities and violent conflict: evidence from cases." *International security* 19, no. 1 (1994): 5-40.

Witmer, Frank DW, Andrew M. Linke, John O'Loughlin, Andrew Gettelman, and Arlene Laing. "Subnational violent conflict forecasts for sub-Saharan Africa, 2015–65, using climate-sensitive models." *Journal of Peace Research* 54, no. 2 (2017): 175-192.

Response 6. Dear reviewer, yes this is our claim. You are right that most analysis is now focused on sub-national level (economic regions, political zones), but very few studies are city/town specific. This is largely because the confounding causal mechanisms of interest are often not available at city or settlement level. However, we are still interested in (for this paper) on whether statistical laws hold and how a prediction algorithm can be developed for city governors. We have included your recommendations above and updated our literature review to reflect this.

Comment 7. When and where terrorists attack are very context-specific behavioral types. Terrorist behavior in Iraq diverges from those in Syria, versus Ukraine, and others do so because such behavior is non-random. These are driven by intra-organizational dynamics (leadership challenge, preventing splits and bolster group identity), external resilience (capacity of state security agencies, terrain, social cohesion and social support for terrorist groups) and ideology of the group (religious, ethno-nationalist etc.) Where and how frequent ISIS attacks in Iraq is different than its

behavior in Syria. There are even further divergences between terrorism in urban areas of failed (failing) states, democracies and authoritarian countries. Trying to impose a general 'supra-level' explanation to these varieties will yield inconclusive results, and the study reports this inconclusivity itself. The study can escape this deadlock by cross-checking with the UCDP/PRIO data (or ACLED) and see whether non-terrorist sub-national violence has any direct correlation with terrorism. This will enable the author to distinguish between areas that are suffering BOTH from high intensity of conflict AND terrorism, and those where the two phenomena are not directly related (high violence, low terrorism or low violence, high terrorism).

Response 7. Dear reviewer, indeed I agree with your view point. I think there is value in getting a supra-level understanding. We have shown that the attacks, even across different urban locations and conflict genres, belong a common random pattern. This could make sense (as it has been shown for many other fields), if the multiple factors that contribute to it are numerous and independently distributed. Take buses arriving in busy cities (which are never on time). This has been shown to follow the same memoryless distribution as terrorism, because the multitude of factors that affect it (traffic, driver behaviour, route, passenger behaviour) are all broadly independent. As a result, the compound effect is a random distribution that is common across all bus arrivals. We have tried to capture the value of our statistical approach in this paper by saying the following in the Introduction:

Statistical analysis of complex processes, even across diverse genres and mechanisms have value in data driven prediction. It has been shown that many complex processes with a multitude of different causal factors can exhibit common statistical patterns that aid prediction, e.g. bus arrival time in busy urban areas.

As such, whilst I totally agree with you that the detailed mechanisms are important and distinguishes violence across genres and mechanisms but having a statistical understanding of the overall pattern is also useful from a data-driven prediction framework perspective.

Comment 8. I think the methodical sophistication of the paper should more directly and robustly interact with the terrorism studies literature, especially with those studies that explore how spatio-temporal dynamics interact with the more relevant and important HUMAN and SOCIAL drivers of terrorism. While I understand the author's models, I'm a social scientist and the models themselves must be reviewed by a reviewer from mathematics, statistics or econometrics fields. But revising the paper in line of above comments will make this a highly citable and useful study for the terrorism literature.

Response 8. I believe Reviewer 2 is a physical science / computer science reviewer, and his/her comments have been addressed below. I have endeavored to ensure that the statistical models are sound, and that there is a relevance and contribution to the humanities and social science research, as you point out.

Reviewer: 2

Comments to the Author(s)

I have read the manuscript RSOS-190645 'Common Statistical Patterns in Urban Terrorism' with much interest and it presents important insights to the relationship between the interval and intensity of terrorist attacks. However, there are some points that, from my perspective, need to be addressed before publication.

General Response. Dear reviewer, thank you for taking the interest and reviewing my manuscript. I have gone through your comments and tried to address the comments. I have highlighted revised new or significantly changed text in blue.

Comment 1) Framing: The framing of the paper could highlight the main contribution more clearly. My reading of the paper is that the findings in regard to the interval between attacks are most central and that the intensity findings (Results a page 3) are more of a side note.

Response 1) Indeed, and we have now sharpened this to say in the Contributions section:

Here, we show that despite diverse conflict genres and multiple confounding mechanisms in play, all global cities suffer attacks describable by a common statistical pattern. The memoryless nature of this pattern suggests that multiple causal mechanisms are independent to each other and that prediction is not helped by the knowledge of previous attacks.

Comment 2) Literature: There is an early literature on the linkage between severity and duration of conflict that might be relevant for this this manuscript to consider. Weiss, 1993; Klingberg 1966; Voevodsky, 1969.

Response 2) Thank you, we have added some of these relevant references to the Introduction along with other references recommended by Reviewer 1. The older studies tend to focus on significant wars across long time scales, as opposed to detailed geo-tagged event data of today.

Comment 3) GTD database. There probably needs to be a short discussion about how terrorism is defined (especially because the scientific and political use of the term can differ) in the context of the GTD database and the potential overlap with events that one could also consider asymmetric warfare (especially events in Iraq and Afghanistan). Also a short note on why the whole GTD is used and not a particular subset (e.g. target type or attack type) would be helpful.

Response 3) Thank you, this is very useful comment and also tangentially pointed out by Reviewer 1. Since post-Cold War, violence between terrorism, politics, criminal enterprise (e.g. narcotics) has become interleaved. Often, trans-national organisations like ISIS participate in all above aspects. As such, studies have shown that it has become difficult to separate the different genres of violence both statistically and contextually \cite{Findley12}. Therefore, it makes sense to consider GTD in its entirety, which is the violence between a non-state actor and other targets (state or non-state). We have clarified this where introduced GTD data and also in discussions.

Comment 4) Page 6 last paragraph. The manuscript claims that 'the distribution given the attack intensity is robust across different urban scales and climates'. It is not clear how this robustness is established.

Response 4) We have added a comment to see the robustness of K-L divergence in Figure 4, which shows information loss from top 50 cities across different geographies, conflict genres, and conflict sizes. Here, we replot Figure 3 for different longitudes, and we can see that the information loss (-1.2 to -2.8 nats) is relatively small across diverse geographies. We have added this plot to the Methods section.

Comment 5) Page 7 first paragraph. The paragraph states that sequential attacks in each city are unrelated and that a 'possible reason is that each terrorist attack depends on a large number of variables (i.e., organization, logistics, finance, personal, evading detection, and opportunity). This raises question about the general modeling approach. Couldn't the relationship between number of attacks and time interval be modelled controlling for a number of factors?

Response 5) If I understand you correctly, you are asking if we can check the contribution of each of the factors? This has indeed been done already extensively in literature. Some factors are latent/hidden, others are not available at a city level granularity – which is why most studies are at national or regional level. This paper tries to find an overall statistical pattern – which is still useful. As discussed with Reviewer 1, whilst I totally agree with you that the detailed mechanisms are important and distinguishes violence across genres and mechanisms but having a statistical understanding of the overall pattern is also useful from a data-driven prediction framework perspective. We discuss this in the paper:

Statistical analysis of complex processes, even across diverse genres and mechanisms have value in data driven prediction. It has been shown that many complex processes with a multitude of different causal factors can exhibit common statistical patterns that aid prediction, e.g. bus arrival time in busy urban areas.

Comment 6) The motivation for using the Kullback-Leibler Divergence to assess model fit/performance is not clear. How is this better than just comparing expected time intervals to true intervals (e.g. RMSE). In addition, it was not clear whether the predictions are assessed within the training sample or whether they were assessed on a test sample. It would be recommendable to assess predictions on a test-sample.

Response 6) Thank you for this comment and it is important to clarify this. We use the R2 value to determine goodness of fit of distribution in the training stage. Compared to RMSE, the K-L divergence tells us how much information is lost in creating a predictive model based on our assumption that it fits a certain exponential distribution. In this case, we lose -1.2 to -2.8 nats of information, which can help us think what factors are in play in future work. The K-L divergence is based on predicted sample (see Figure 3 title).

Comment 7) The motivation to use the spectrogram analysis could be made clearer on page 3 sixth paragraph. The main finding is that the growth in death-tolls from terrorist attacks is 'due to an increase in the number of slower and bigger casualty attacks'. It would be good to explore or at least discuss to which extent this might be due to coding bias in GTD (see Iraq/Afghanistan and temporal trends in the data.).

Response 7) Thank you for this comment. The spectrogram is indeed motivated by the need to see if small-scale attacks or large-scale attacks are contributing towards the trend. We know that statistical certainty of large-scale attacks is lower (sparse data). Whether the reason is down to GTD coding, we offer a discussion as follows in our spectrogram section:

The spectrogram trend we see is not universal, but the insight is generalizable. That is to say, an abundance of small casualty attacks gives our prediction greater accuracy (less information loss), whereas fewer high casualty attacks leads to a poor estimation of the statistical parameter(s) used for prediction – leading to growing uncertainty in predictions. Specifically, we see greater high casualty attacks in Iraq and Afghanistan, but not so in Colombia and Mexico.